# Problematic Imaging Diagnostics of Musculoskeletal Gossypiboma with Chronic Expanding Hematoma Mimicking Malignant Lesion

**DOI:** 10.3390/diagnostics13091592

**Published:** 2023-04-29

**Authors:** Tomas Kucera, Libor Prokes, Jiri Soukup, Jindra Brtkova, Ondrej Valtr, Pavel Sponer

**Affiliations:** 1Department of Orthopedic Surgery, University Hospital in Hradec Kralove, Sokolska 581, 500 05 Hradec Kralove, Czech Republic; 2Charles University, Faculty of Medicine in Hradec Kralove, Simkova 870, 500 03 Hradec Kralove, Czech Republic; 3The Fingerland Department of Pathology, University Hospital in Hradec Kralove, Sokolska 581, 500 05 Hradec Kralove, Czech Republic; 4Department of Radiology, University Hospital in Hradec Kralove, Sokolska 581, 500 05 Hradec Kralove, Czech Republic

**Keywords:** musculoskeletal imaging, gossypiboma, chronic expanding hematoma, malignant tumor, autologous bone grafts

## Abstract

Both musculoskeletal gossypibomas and chronic expanding hematomas have been rarely reported; the reports that do exist are usually case reports. Our objective is to demonstrate problematic imaging diagnostics of an unusual presentation mimicking a malignant lesion. We report the case of a 47-year-old man who underwent bone graft harvesting from the iliac crest for spinal fusion due to scoliosis at 18 years of age, and 29 years later, he developed a growing, painful tumor at the original donor site (a bone defect in the iliac crest). It was challenging to differentiate a hematoma from a malignant tumor based solely on clinical and radiological workup, including an ultrasound-guided needle biopsy focused on viable tissue. The definitive diagnosis of a gossypiboma with a chronic expanding hematoma was based on histopathological assessment after wide surgical resection—a chronic expanding hematoma with multiple foamy macrophages and giant cells engulfing foreign material (original surgical hemostatic sponge).

A 47-year-old man presented to our outpatient department with a growing, painful tumor in the region of the left iliac bone. He sustained bone graft harvesting from the iliac crest for spinal fusion due to scoliosis 29 years ago, and he had not had any problems in the past. Therefore, no imaging method was available for comparison. He stated he did not have any history of trauma, and he had observed the tumor for one year. The patient had already undergone an inconclusive diagnostic checkup at a different hospital, with an open biopsy only yielding nonspecific findings of tissue necrosis. Imaging methods were used (Figure 1).

The combination of aggressive bone destruction, lesion extending into the pelvis, and a non-aggressive sclerotic margin with a narrow zone of reactive osteosclerosis were considered controversial. A significant portion of the lesion infiltrated the gluteal muscles, displaying subtotal central necrosis and a thin viable (enhancing) rim preserved in the periphery. Calcification of an unusual morphology was visible in the center of the lesion. Bone destruction associated with soft tissue mass and an irregular tumor shape suggested a malignant lesion. Ultrasound assessment and ultrasound-guided needle biopsy followed (Figure 2).

The histological findings from the ultrasound-guided needle biopsy were considered nonspecific and inconclusive. No tumor was identified, and a recommendation was made to exclude histiocytosis or Erdheim–Chester disease (Figure 3).

These diagnoses were subsequently excluded based on a hematologic workup and a bone scan. At angiography, the mass was hypovascularized and thus unsuitable for therapeutic embolization. We therefore decided to perform a wide surgical excision and histologic assessment of the whole mass. We performed an incision along the iliac crest. Beneath the gluteal muscle, we found a tumorous mass 10 × 10 cm adjacent to the original site of bone graft harvesting. It consisted of a fibrous pseudocapsule, an old hematoma, and necrotic tissue. After wide resection, we sent this mass for histologic assessment. We found remnants of a foreign material near the iliac bone similar to a textile (Figure 4).

The donor site of the iliac crest had smooth margins without macroscopic changes from recent destruction. The procedure revealed that the iliac bone defect was the original donor site, not new osteolytic destruction. Finally, we refixed the gluteal muscle by via a transosseous suture to the iliac crest. We sent the tumorous mass for histopathological assessment (Figure 5).

The final diagnosis corresponded to a gossypiboma with a chronic expanding hematoma.

Non-weight-bearing with crutches was recommended after surgery for six weeks due to the healing of the refixed gluteal muscles. We have not observed any complications or relapses during follow-up for five years.

A chronic expanding hematoma is a rare entity resulting from trauma, surgery, and bleeding disorders and has been mentioned infrequently in the literature with various presentations [1,2,3]. It is a mixture of old and new blood with time-related changes present, accompanied by necrotic degradation and liquefaction and a fibrous pseudocapsule [4]. Additionally, our needle biopsy revealed a rigid capsule. The histological features are reportedly a mixture of blood breakdown products, granulation tissue with capillary ingrowth, and inflammatory tissue [5]. It usually lays dormant for many months before suddenly expanding, similar to a chronic subdural hematoma. In our case, the hematoma started to expand 29 years after the original surgery.

Similarly, Sakamoto and Matsuda described a chronic expanding hematoma as a late complication from 24 to 45 years after thoracoplasty [6]. The expansion is thought to be related to the inflammation stimulated by blood breakdown [7]. These cellular breakdown products then cause a fibroblastic reaction. Inflammation increases vascular wall permeability and bleeding from dilated capillaries into the granulation tissue beneath the capsular wall, resulting in the expansion of the hematoma [8]. It is complicated to differentiate a chronic expanding hematoma from a malignant tumor, as both may slowly expand and occasionally erode the bone [9]. Magnetic resonance imaging is considered an excellent diagnostic modality. The findings of central heterogeneity on both T1-weighted and T2-weighted images with a peripheral rim of low signal intensity reflect fluid collection of fresh and altered blood, and a pseudocapsule (fibrous wall) with low T1- and T2- signal intensity is reportedly more consistent with a hematoma than a malignancy. Unfortunately, this fibrous hypointense pseudocapsule can be confused with thinned expanded remnants of cortical bone. This combination of imaging can be considered characteristic for diagnosing a chronic hematoma [10]. A gossypiboma develops due to a reaction to a retained surgical sponge, sometimes many years after the initial surgery [11]. The retained surgical sponge triggers two biological responses: an aseptic fibrinous response due to foreign body granuloma or an exudative reaction, leading to abscess formation [12]. The exudative response is more common, leading to abscess formation with or without secondary bacterial contamination. The aseptic fibrous inflammatory reaction is usually rare, developing into cyst formation with adhesion to the surrounding structures. This type of presentation could be latent for many years [13]. Only 6% are reported after orthopedic surgery, compared to 52% following abdominal surgery, and no fatal complications have been reported in musculoskeletal sites [14]. In our case, it was problematic to differentiate a musculoskeletal gossypiboma with a chronic expanding hematoma from a malignant tumor based solely on clinical and radiological workups. The definitive diagnosis was based on histopathological assessment after the wide surgical resection.

In conclusion, both musculoskeletal gossypibomas and chronic expanding hematomas have been rarely reported; they can develop for months, but sometimes with a latency of decades; they can also mimic a malignant lesion. The consideration of these subjects in differential diagnosis and the importance of bringing them to medical attention should be emphasized.

## Figures and Tables

**Figure 1 diagnostics-13-01592-f001:**
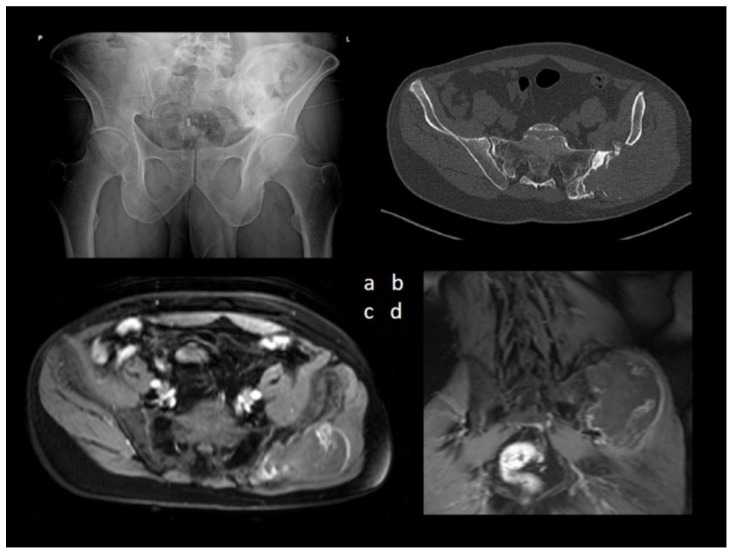
(**a**) Anteroposterior radiography of the pelvis revealed an extensive mixed lytic and sclerotic lesion of the left iliac bone. (**b**) Axial CT image of the lesion and MRI T1-weighted fat-suppressed contrast-enhanced sequences of the pelvis—(**c**) axial view and (**d**) coronal view demonstrated a large, aggressive lesion causing extensive osteolysis of the left iliac bone.

**Figure 2 diagnostics-13-01592-f002:**
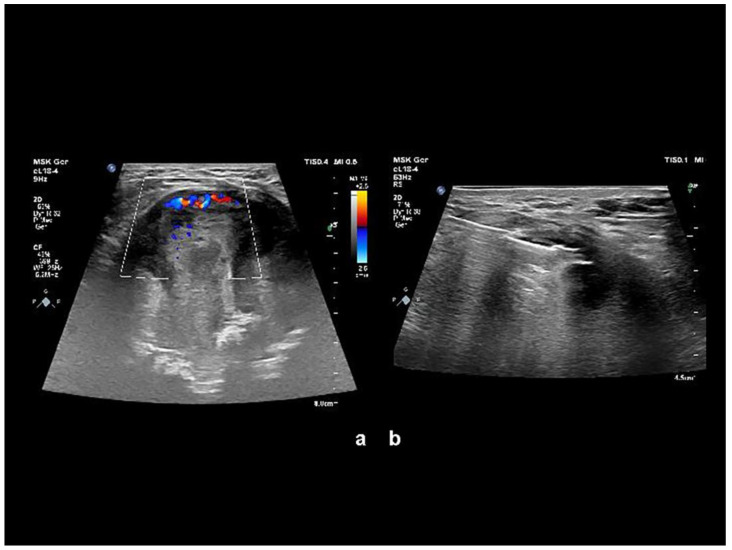
(**a**) Color Doppler imaging sonography revealed inhomogeneous expansion with alternating major and minor hyperechogenic areas, with portions significantly hypervascularized and completely without hypervascularization. (**b**) Ultrasound-guided needle biopsy focused on viable tissue was performed. At the puncture, the expansion capsule was rigid and “fibrous”.

**Figure 3 diagnostics-13-01592-f003:**
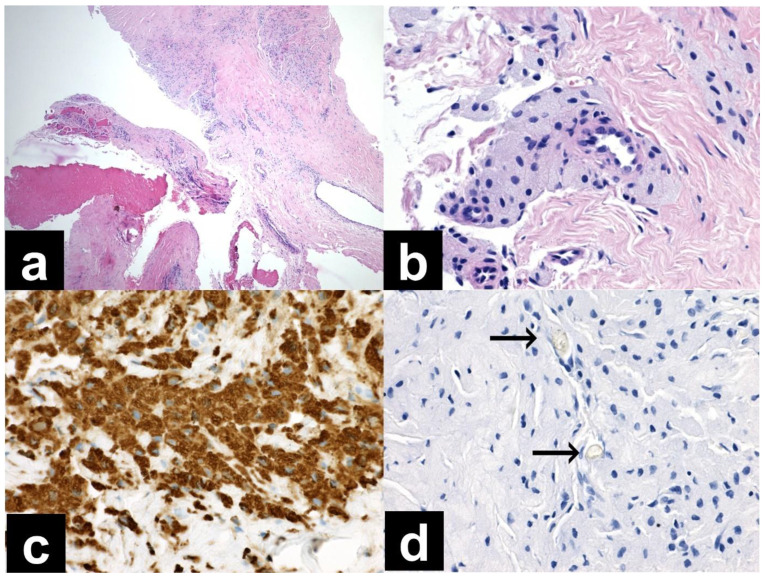
(**a**) Needle biopsy, dense collagen tissue, and brightly eosinophilic material consistent with older hemorrhage or necrotic tissue (40× magnification, H&E), (**b**) needle biopsy, aggregates of foamy macrophages surrounding a small vessel (400× magnification, H&E), (**c**) needle biopsy, strong immunohistochemical positivity of CD68 in macrophages (400× magnification), (**d**) needle biopsy, negative PAS reaction in two foreign bodies (arrows, 400× magnification).

**Figure 4 diagnostics-13-01592-f004:**
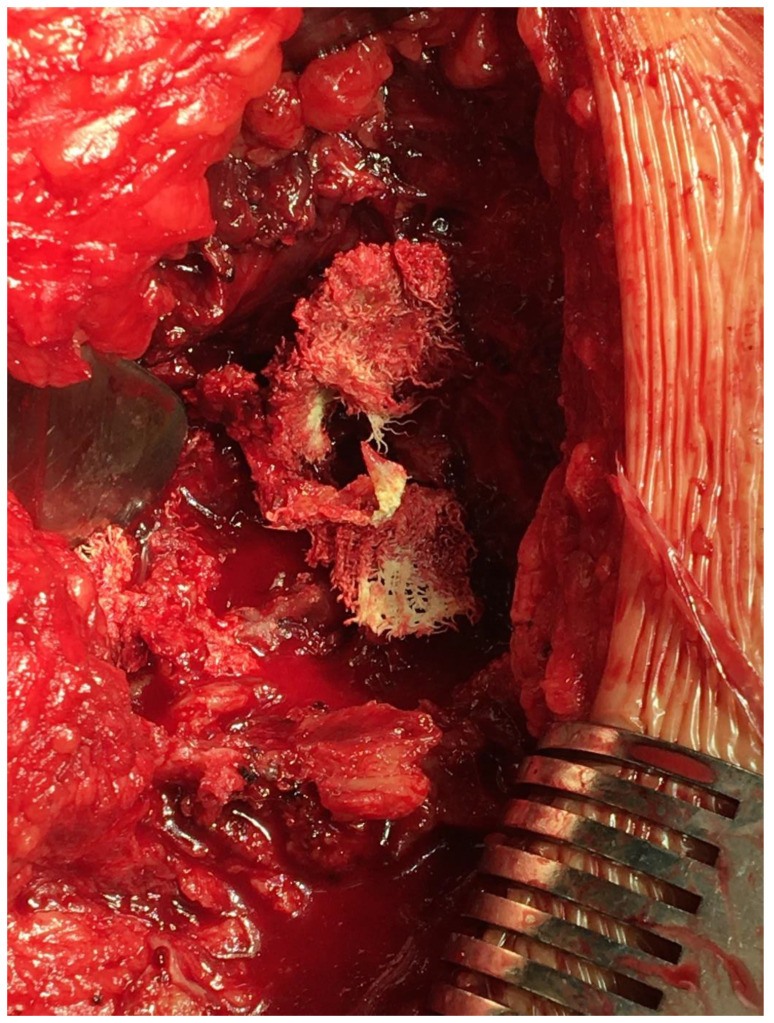
Remnants of the original surgical hemostatic sponge and old hematoma near the iliac bone defect.

**Figure 5 diagnostics-13-01592-f005:**
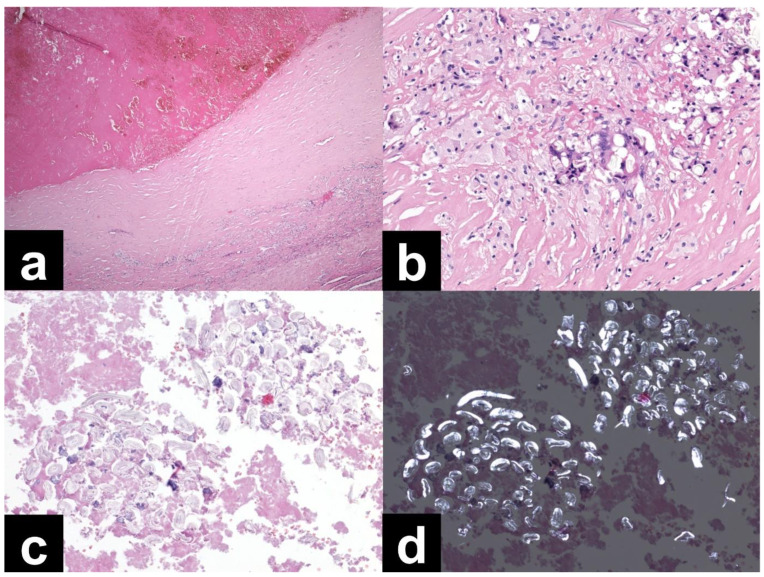
The tissues were mainly old hematoma with vascular channels, rimmed by scar tissue with multiple foamy macrophages and giant cells engulfing foreign material. The foreign material was identifiable in the blood clots and consisted of fibrillary structures arranged into clusters. The material was polarizable, and overall morphology was compatible with a piece of textile, possibly a surgical sponge. (**a**) Resection specimen; most of the sample was composed of older hemorrhage (upper half) surrounded by a dense collagenous capsule (40× magnification, H&E). (**b**) Resection specimen; multiple foamy macrophages and giant cells engulfing foreign bodies in the collagenous capsule (200× magnification, H&E). (**c**) Resection specimen; foreign material composed of fibrillary structures arranged in arrays, surrounded by blood clots (200× magnification, H&E). (**d**) Resection specimen; the same area from 2C with foreign material showing polarization (200× magnification, H&E, polarized light).

## Data Availability

The raw data supporting the conclusions of this article will be made available by the authors, without undue reservation.

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
