# Peer review of "Problematic Imaging Diagnostics of Musculoskeletal Gossypiboma with Chronic Expanding Hematoma Mimicking Malignant Lesion"

_diagnostics, 2023, doi:10.3390/diagnostics13091592_

Round 1
Reviewer 1 Report
This article is dedicated to an important problem of interventional medicine, mainly intra-cranian, intraabdominal, or intrathoracic surgery. Gossypiboma or textiloma, the expression of a foreign-body related inflammatory pseudotumor arising from retained non-absorbable cotton matrix inadvertently left behind during surgery, developed during months ,but sometimes with a latency of decades, occurs with a frequency of 1 to 1,000-1,500 interventions, mostly in emergency and laparoscopic procedures. Although relatively rare, delayed- or misdiagnosed, it can increase the severity of evolution and even be the cause of mortality; it accounts for about 50% of malpraxis claims for retained foreign bodies, an important burden for physicians. Therefore I consider important to bring this subject into medical attention.
The article presents a pictorial view of the investigations performed for the diagnosis of a gossypiboma ( radiography, CT and MRI, color Doppler imaging sonography, angiography, finalized with histopathological assessment after a wide surgical resection), followed by discussion.
The article is worth to be published underlining the challenges and threats of this disorder.
Comments.
In order to learn from your presentation, besides the interesting and informative images, I would propose:
1.-a short introduction, followed by a short case report revealing the long latency (29 years ) of this gossypiboma, and after the presentation of images and discussions ,at the end of the article, to present the conclusions with take-home messages
2.-under the figures I would consider to explain the data connected to the investigation performed and to describe the images of the obtained results, but the clinical history and data should be presented in the case report ( figure1,5 )
Author Response
Dear Sir/Madam,
Thank you very much for your review. Following submitting the manuscript, an assistant editor suggested changing "Case report "to "Interesting Images. "I changed it according to the "Instructions for Authors "- Interesting Images: The number of images is at the author's discretion. No regular manuscript text (introduction/methods/results/discussion) should be included. Instead, images should be accompanied by detailed legends with no restriction in length.
However, according to the requirements of all reviewers, I rearranged the text and added a conclusion with the take-home message.
Best wishes,
Tomas Kucera

Reviewer 2 Report
The manuscript "Problematic Imaging Diagnostics of Musculoskeletal Gossypiboma with Chronic Expanding Hematoma Mimicking Malignant Lesion" only shows figures with comments. There is no paper. Such a kind of manuscript is not worthy of publication.
Author Response

(The authors gave the same response as above.)

Reviewer 3 Report
It is well presented case report although not rare yet it is rarely reported
it is good to consider always in DD
very few grammar and spelling errors
explain on the figure that can be written in the case report and let the photo tags to be kept to minimum words that represent the photo only
no formal case report sections case report history ,examination ,radiology and management then discussion and conclusion please rearrange
no cnclusion
Author Response

(The authors gave the same response as above.)

Round 2
Reviewer 1 Report
Comments
-the present article is worth to be published bringing to the attention of practitioners a subject, revealing a rare surgery complication with diagnostics challenges
-in my opinion, the present output is suitable
Corrections
not to repeat the same thing in two successive sentences (abstract): Our objective is to demonstrate problematic imaging diagnostics of an unusual presentation of the complication following bone graft harvesting from the iliac crest for spinal fusion mimicking a malignant lesion.
-delete these lines, as they are already introduced in the text (page 8): Figure 1The combination of aggressive bone destruction, lesion extending into the pelvis, and a non-aggressive sclerotic margin with a narrow zone of reactive osteosclerosis were considered controversial. A significant portion of the lesion infiltrates the gluteal muscles, displaying subtotal central necrosis and a thin viable (enhancing) rim preserved in the periphery. A calcification of an unusual morphology is visible in the center of the lesion Figure 4Funding: This work was supported by the Cooperatio Program (research area SURG) and MHCZ – DRO (UHHK, 00179906), and from European Regional Development Fund-Project BBMRI-CZ: Biobank network – a versatile platform for the research of the etiopathogenesis of diseases, No: EF16 013/0001674.
-it is a long phrase with three predicates on page 8; split it into three sentences separated by a semicolon:
In conclusion, both musculoskeletal gossypiboma and chronic expanding hematoma have been rarely reported; they can develop for months, but sometimes with a latency of decades; they can also mimic a malignant lesion. The consideration of these subjects in differential diagnosis and the importance of bringing them to medical attention should be emphasized.
Author Response
Dear reviewer,
Thank you very much for your valuable comments, I have followed your recommendation with one exception, due to the customs of our institution I had to leave the Funding.
Best wishes,
Tomas Kucera

Reviewer 2 Report
The manuscript is still not suitable for publication. It is just a collection of images and the journal template has not been used.
Author Response
Dear reviewer,
I'm sorry, but I would like to ask you for your specific requests for editing the manuscript. I have corrected it according to the requirements of the other two reviewers, the format is "Interesting Images" according to the template of the journal, i.e. as you write a collection of images, they have no further requirements. I will of course take your advice. I very much appreciate your help.
Best wishes,
Tomas Kucera